# Idiopathic Abdominal Wall Endometrioma: Case Report with Investigation of the Pathological, Molecular Cytogenetic and Cell Growth Features In Vitro

**DOI:** 10.3390/ijms26020775

**Published:** 2025-01-17

**Authors:** Jean Gogusev, Yves Lepelletier, Henri Cohen, Olivier Ami, Pierre Validire

**Affiliations:** 1Université Paris Cité, Institut Cochin, INSERM UMR 1016, CNRS 8104, 22 Rue Méchain, 75014 Paris, France; 2W-MedPhys, 128 Rue la Boétie, 75008 Paris, France; y.lepelletier@gmail.com; 3Université Paris Cité, Institut Imagine, 24 Boulevard Montparnasse, 75015 Paris, France; 4INSERM UMR 1163, Laboratory of Cellular and Molecular Basis of Normal Hematopoiesis and Hematological Disorders: Therapeutical Implications, 24 Boulevard Montparnasse, 75015 Paris, France; 5Service de Chirurgie Gynécologique, Institut Mutualiste Montsouris, 42 Bd Jourdan, 75014 Paris, France; henricohen@imm.fr; 6Ramsay Générale de Santé, Clinique de la Muette, 46-48 rue Nicolo, 75116 Paris, France; dr.olivierami@gmail.com; 7Service d’Anatomie Pathologique, Institut Mutualiste Montsouris, 42 Bd Jourdan, 75014 Paris, France; pierre.validire@orange.fr

**Keywords:** endometriosis, abdominal wall endometrioma, cell culture, karyotype, FISH analysis, c-MYC proto-oncogene

## Abstract

Abdominal wall endometriosis (AWE) is a clinical disorder with unknown pathogenesis with an incidence between 0.03% and 1% in women affected by cutaneous/scar endometriosis. We investigated the pathological, molecular cytogenetic and cell proliferation features of a primary AWE developed in rectus abdominis muscle in a patient without co-existing pelvic endometriosis. An investigational model of cultured stromal cells was additionally established. Histologically, the lesion revealed areas of endometrial-like glands surrounded by a thick stromal layer in addition to numerous disseminated foci composed exclusively of stromal cells. Beyond the strong expression of Estrogen (ER) and Progesterone receptors (PRs), consistent immunolabeling for several mesenchymal stromal/stem cell antigens and oncoproteins was revealed in both the endometrioma as well as in the cultured stromal cells. The Fluorescence in situ hybridization (FISH) analysis of the endometrioma demonstrated a structural alteration of the c-MYC protooncogene, with a mean of three gene copies in 3% to 5% of both glandular and stromal cells. The FISH assay applied on the cultured cells showed c-MYC gene amplification, with an average number of more than six gene copies in 18% to 25% of the cellular nuclei. Altogether, these results markedly highlight the pathological and molecular features of idiopathic AWE essential for histo-pathogenetic categorization.

## 1. Introduction

Endometriosis in women is defined by the presence of endometrial glands and stroma outside the uterine cavity. Abdominal wall endometriosis (AWE) variety is classified as primary and secondary, with the primary form is characterized by the presence of functional endometrium in the skin, representing about 0.4% to 4.0% of all forms of endometriosis and accounts for 30% to 40% of cases with cutaneous/scar endometriosis [1,2,3]. In this context, the most frequent abdominal location is the umbilicus, being a predilection site for ectopic endometrial implants [4].

The primary idiopathic AWE is a rare clinical entity that is not related to lesions representative of internal or external genital endometriosis [3,5,6]. From a review of 445 published cases in the literature, Horton et al. [7] determined that primary abdominal endometriosis develops in 20% of the studied subjects as idiopathic and is not interconnected to the form appearing after the surgical scars [7]. Comparatively, the secondary abdominal endometriosis form is more frequent, and its origin is fundamentally explained by the iatrogenic dissemination of the eutopic endometrial cells after surgical interventions such as cesarean section, laparoscopy, hysterectomy, episiotomy and amniocentesis [2,8,9].

The predominant theory proposed to explain the pathogenesis of primary abdominal wall endometriosis suggests the migration of endometrial cells to the umbilicus through the abdominal cavity or via the lymphatic system [9,10,11,12]. The metaplastic theories propose that the origin of idiopathic AWE is either due to changes of the primitive pluripotential mesenchymal cells situated in the coelomic cavity [1,12] or the metaplastic alteration of the embryonic epithelial cells present in urachal remnants [13]. Other pathogenetic concepts emphasize the activation of a local inflammatory process due to growth factor anomalies [14,15], estrogen stimulation through estrogen-binding receptors [16] and potential epigenetic changes [17].

In the present study, we describe the histopathology, immuno-histochemical features, molecular cytogenetics and cell growth characteristics of an idiopathic primary AWE developed in a patient without concomitant pelvic endometriosis. The lesion exhibited a specific immunohistochemical profile, a propensity of the stromal component to grow in long-term cell culture and molecular alterations of the c-MYC gene expressed as copy number gain. Overall, the presented data reveal previously undescribed pathological and molecular features involved in the development of idiopathic AWE and highlight substantial cellular parameters that underscore the differences between primary and secondary endometriotic forms.

## 2. Results

### 2.1. Histopathological Analysis

Microscopically, the endometrioma lesion exhibited nodular structures containing distorted endometrial-like glands surrounded by a large stromal component implanted in the fibromuscular tissue (Figure 1A). Additionally, the focal glandular/stromal areas of the lesion were surrounded by numerous dispersed foci composed of exclusively stromal cells admixed with larger epithelioid-like cells that were observed through the entire lesion (Figure 1B).

### 2.2. Stromal Cell Culture

On phase contrast microscopy, the primary culture showed numerous adherent cell aggregates growing as an admixture of polygonal epithelial and myofibroblast-like cells, which proliferated mostly as a monolayer. After the early passages, the cells appeared rather polygonal and irregularly shaped with ovoid nuclei and abundant cytoplasm and showed cytoplasmic extensions (Figure 1C). Detached from the culture flask and prepared on slides, the cells changed into a round morphology appropriate for the cytochemical and FISH analyses (Figure 1D). After several passages, the cells started continuous growth as elongated, large mesenchymal-like cells with a doubling time between 5 and 6 days. 

### 2.3. Immunohistochemistry

The immunohistochemical characteristics of the abdominal wall endometrioma are shown in Figure 2 and Table 1. Consistent nuclear expression of the estrogen receptor (ER) was detected in the glandular lining cells, while stromal cells are moderately stained (Figure 2A). In contrast, strong expression of the progesterone receptor (PR) was present in the stromal component (Figure 2B) compared to PR immunostaining of the glandular cells (Table 1). The Ki-67 nuclear protein was detected in approximately 15% of the cells in the stromal areas, indicating proliferative activity (Figure 2C, Table 1). Concerning the c-Myc immunoreactivity, the stromal cells exhibited intense nuclear staining (Figure 2D), while weak to moderate and heterogeneous labeling was observed in the glandular cells (Figure 2D and Table 1). Strong c-MYC nuclear staining was also present in the cells from the scattered stromal foci (Table 1). The expression of p16(INK4a)-related molecules was detected predominantly in the stromal cell layer (Figure 2E). Comparatively, less abundant nuclear p53 immunolabeling was observed in both the gland lining epithelial cells and the stromal component (Figure 2F). Regarding the expression of mesenchymal stromal cell antigens, abundant immunolabeling with the anti-CD10 membrane endopeptidase antibody was observed in the stromal cell layers as well as in the isolated stromal foci (Figure 2G, Table 1). The anti-CD73 antibody strongly stained only the stromal cell layer as well as the cells constituting the isolated foci (Figure 2H, Table 1). Likewise, the anti-CD90 and CD105 antibodies stained only the endometrioma stromal cell component (Table 1). Comparatively, consistent immunoreactivity for the endometriotic marker protein WT1 was detected in nearly 50% of the glandular epithelial cells and in 45% of the cells in the stromal component (Table 1).

### 2.4. Immunocytochemistry

The immunocytochemical traits of the cultured stromal cells are presented in Figure 3 and Table 1. While 94% of the stromal cells exhibited intense membrane and cytoplasmic staining for the vimentin protein (Figure 3A), they were not immunolabeled for cytokeratin (Table 1). Consistent nuclear staining for c-MYC was present in more than 70% of the proliferating cells (Figure 3B), while p16 nuclear expression was observed in 40% of them. (Figure 3C). Intense immunostaining for the WT1 protein marker was found in 65% of the cells (Figure 3D), for p53 in 32% (Figure 3E) and for the retinoblastoma tumor suppressor protein (Rb) in 45% of them (Figure 3F). Moreover, the cultured cells were consistently immunolabeled for CD9 (55%), CD10 (85%), CD13 (80%), CD73 (75%), CD90 (68%) and CD105 (61%) mesenchymal stromal cell markers (Table 1).

### 2.5. Cytogenetic Analysis of the Stromal Cells

On RGH banding, the cultured stromal cells examined at two different passages (4th and 17th) demonstrated predominantly normal cell metaphases with two XX female chromosomes (Figure 4). Significantly, the q24 arm of chromosome 8, the site of the c-MYC gene, showed a normal cytogenetic structure.

### 2.6. Assessment of the c-MYC Gene Structure by Fluorescence In Situ Hybridization (FISH)

The FISH analysis applied to the endometrioma sections presented a heterogeneous distribution of the C-MYC gene signals in the interphase nuclei that appeared either as large dispersed or grouped clusters. Amplification of the c-MYC gene, with a mean of three gene signals per nucleus, was observed in approximately 3–5% of both glandular and stromal cells through the lesion (Figure 5A). Nuclei with more than six c-MYC/CEN8 signals were observed in 1–3% (mean value) of both glandular and stromal cells constituting the endometrioma tissue (Figure 5B). FISH hybridization of the cell cultures from different passages with the c-MYC probe revealed a strong amplification, with a mean of six c-MYC clustered signals per nucleus in more than 20% of the cells (Figure 5C). However, a normal female karyotype, including a normal chromosome 8 where the C-MYC gene is located, was revealed by the conventional cytogenetic analysis (Figure 4). In the internal FISH control experiments, hybridization with the c-MYC probe of both normal endometrial tissue sections and peripheral blood lymphocytes showed 100% of the cells with two normal c-MYC fluorescent signals.

## 3. Discussion

At present, mostly descriptive clinical and pathological reports with limited pathogenetic significance are published considering primary and secondary abdominal endometriosis [1,2,3,9,19]. Away from the notions of the clinical distinctiveness between primary and secondary abdominal endometriosis, the occurrence of spontaneous idiopathic AWE form without coexistent pelvic endometriotic lesions remains a disorder with enigmatic pathogenesis [3,9]. So far, no specific histopathological, cytological or molecular features are recognized allowing differentiation of primary abdominal wall endometriosis from the disease arising in abdominal surgical scars, generally described as cutaneous surgical endometriosis [2,9]. Accordingly, the rarity of AWE disease and the lack of suitable experimental models hamper original investigations to categorize the pathological and/or molecular features of abdominal endometriosis forms.

The presented results emphasize that, in addition to the strong expression of estrogen and progesterone hormonal receptors, several proto-oncogenes and mesenchymal stromal/stem cell (MSC) proteins are steadily expressed in the AWE lesion as well as in the cultured stromal cells. In addition to the strong expression of the c-MYC oncoprotein, a structural aberration of the c-MYC gene was observed by FISH analysis, expressed as a copy number gain in 3% to 5% of both glandular and stromal cells through the endometrioma tissue and in more than 20% of the cultured stromal cells from the lesion. In this context, aberrant expression of c-MYC, p53, p16 and Rb oncoproteins was previously reported and related to the impaired apoptotic process involved in the development and progression of the various endometriotic forms [20,21,22,23,24,25,26,27]. Also, the enhanced expression of c-MYC in ectopic endometrium was previously shown in connection with the epithelial to mesenchymal transition process, as an important prerequisite for the development of endometriotic lesions [21,28,29]. Overall, our data accentuate that the c-MYC proto-oncogene plays an important mechanistic role in the development and progression of idiopathic primary abdominal wall endometriosis. Likewise, aberrant expression of c-MYC, p16 and p53 molecules was reported to be associated with the high cellular proliferation rate observed in ovarian neoplastic processes [21,29,30]. Furthermore, the presented findings indicate consistent expression of several specific mesenchymal stromal markers in both AWE and cultured stromal cells. Remarkably, strong expression of CD73, CD90 and CD105 antigens was previously described in pluripotential mesenchymal stromal cell lineages obtained from both ectopic endometrial tissue and from bone marrow [31,32,33]. In the same context, investigations of the mechanisms controlling the development of idiopathic primary endometriosis from distant locations suggested that this phenomenon may be an independent pathogenetic process, initiated by locally committed pluripotent mesenchymal stem cells expressing CD73, CD90 and CD105 MSCs molecules [34]. Similarly, two different malignancies arising in cutaneous umbilical endometriomas, histologically described as clear cell and serous adenocarcinomas, were linked to altered proto-oncogene structures and aberrant mesenchymal stromal cell protein expression [35,36,37]. In the same context, several studies have reported an epidemiologic and histopathological association between endometriosis and ovarian neoplastic processes. Yet, the molecular and cellular mechanisms remain not well understood. In this regard, the genetic link has been suggested through mutations in PIK3CA, KRAS, ARIDIA1A and other genes [38]. Also, little is known concerning the correlation between the implantation sites and endometrial-related carcinoma. Future studies are needed to establish or not the link between these pathologies. Moreover, in contrast with the benign abdominal scar endometriomas, these neoplastic proliferations are not immunoreactive for estrogen and progesterone receptors, but they strongly express the p16, p53 and WT1 oncoproteins [37].

In conclusion, the presented immunohistochemical, molecular cytogenetic and cultured cell profile analyses suggest particular immuno-phenotypic and cell growth features of AWE developed in the rectus abdominis muscle in women not affected by pelvic endometriosis. Altogether, the increased expression of proteins regulating the apoptosis/necrosis process and the consistent expression of particular mesenchymal stromal/stem cell proteins markedly contribute to distinguishing between primary and secondary AWE forms and bring novel information regarding this disease pathogenesis.

## 4. Materials and Methods

### 4.1. Patient

A 38-year-old nulliparous woman with normal menses clinically presented with a large nodule deeply implanted in the rectus abdominis muscle without history of abdominal surgery. The physical examination on admission revealed a large, solid nodular mass, 5 × 6 cm in diameter, deeply implanted in the abdominal rectus muscle, sideways to the umbilicus. Previous clinical and imaging investigations did not reveal any concomitant pelvic endometriotic lesions. Additional imaging studies were performed tp distinguish for differential diagnosis with keloid, sister Mary Joseph’s nodule, or other pathology [39]. Abdominal wall ultrasonography revealed the presence of non-cystically altered mass with homogeneous echogenicity and increased vascularity. A subsequent MRI confirmed the presence of a large lump located in the abdominal rectus muscle without connection to abdominal organs. The blood picture and the tuberculosis workup showed no anomalies, and surgical intervention was performed. A large surgical excision of the nodule was carried out, with a progressive dissection around the lump until normal muscle tissue was visible at the margins. Upon discharge, given the fact that the nodular mass was totally excised, no further medical therapy was proposed to the patient. During regular follow-up, two years after the excision, the patient remained free of disease without recurrence. The authors followed the care protocol guidelines approved by the ethics committee in our institution and further obtained informed consent for patient privacy protection.

### 4.2. Tissue Processing and Cell Culture

The surgical excision revealed a large solid nodule, 5 × 6 cm in diameter, which was processed for both routine histopathological analysis and for the in vitro experimental work. The institutional ethics committee approved this investigation, and patient consent was obtained before tissue collection. The majority of the excised mass was fixed in buffered formalin and processed for histology following the standard procedures. Serial tissue sections (4 µm) were made for histopathological analysis and evaluation of the immunohistochemical parameters. For the cell culture, a solid tissue fragment (2 × 1 cm in diameter) was dissected under sterile conditions, and the cell culture procedure performed as previously described [40]. Under phase microscopy, rapidly growing adherent and elongated cells developed after 4–5 weeks in culture that were serially passaged at a density of 2 × 10^5^ cells/mL. The cell cultures multiplied continuously over two years and were phenotypically and cytogenetically characterized. 

### 4.3. Immuno-Histochemistry

Serial sections of the paraffin-embedded endometrioma tissue were deparaffinized, dehydrated in graded ethanol and heated at 65 °C for antigen retrieval before treatment with the specific antibodies. The immunostaining reaction was assessed after incubation of tissue sections with a panel of specific monoclonal or polyclonal antibodies at appropriate dilution (Table 1). The immunoreactivity was revealed with either rabbit anti-mouse or rabbit anti-goat IgG peroxidase-conjugated antibodies (LSAB2 system, Dako, Glostrup, Denmark) according to the provider’s instructions. The immunohistochemical evaluation was performed using the histochemical score (H-score) to assess the intensity of staining and the percentage of stained cells after immunolabeling [41].

### 4.4. Immuno-Cytochemistry

The cells were either grown as adherent in cell chambers (Lab Tek, Fisher Scientific, Illkirch, France) or prepared as cytospins after centrifugation of the cellular suspension onto slides. Afterwards, the slides were washed with PBS buffer, fixed in cold acetone, rinsed again in PBS and incubated with the primary antibodies. The control slides for either tissue sections or cultured cells were incubated with normal non-immune IgG to replace the specific antibodies. Immunoreactivity was evaluated semi-quantitatively as follows: strong staining (presence of more than 65% of stained cells); moderate staining (30% of stained cells); weak staining (±10% of labelled cells) and negative staining not significantly greater than that of the control slides. The percentage of immunoreactive cells for each antibody was determined by counting 500 cells and evaluated by two different observers.

### 4.5. Conventional Cytogenetics Analysis

The cultured stromal cell metaphases from two different passages (4th and the 17th passage) were R-banded (RGH-banding) according to standard protocols as described Rooney [18]. Between 20–30 selected metaphases were counted and 15 of them were photographed and analyzed.

### 4.6. Assessment of the c-MYC Gene Structure by Fluorescence In Situ Hybridization

Serial sections were prepared from the endometrioma tissue and mounted on sialinized slides. The sections were deparaffinized, treated with protease and washed on half-automated VP2000 processor system (Abbott Molecular, Rungis, France). Thereafter, the slides were denatured for 5 min at 75 °C and hybridized at 37 °C overnight in the presence of 10 microliter of MYC/CEN8p probe (Abnova, cat N° FG0009, Walnut, CA, USA). In parallel, the FISH assay was performed on cultured stromal cells previously washed in PBS and fixed in methanol/acetic acid as described by Knoll and Lichter [42]. Post-hybridization washes with appropriate dilutions of SSC buffer were performed at 72 °C and the slides with tissue sections or cells counterstained with 4.6-diamidino-2-Phenylindole (DAPI) (Vector Laboratories, Peterborough, UK). The hybridized slides were then scanned for fluorescent amplification spots by a fluorescent microscope Olympus BX41, using ×40 or ×63 objectives, with a double filter FITC/TRITC) and a capture and image analysis system Cytovision 7.2, spot counting system (Leica Microsystems, Nanterre, France). A total of 100 nuclei were individually evaluated with the ×63 objective by counting red (c-MYC) signals, known to be located on the q24 arm of chromosome 8, and green centromere 8 (CEN8) signals. The c-MYC/CEN8 ratio was calculated, where the signal copy number of c-MYC was divided by the number of chromosome 8 centromere signal. Tissue or cellular nuclei exhibiting three or four copies of the specific signals were considered to have an amplification or gene copy gain. As a negative control, the c-MYC probe was hybridized with sections of normal endometrial tissue biopsies and peripheral blood lymphocytes.

### 4.7. Statistical Analysis

The chi-squared test was used for the statistical evaluation of the FISH analyses. The significance level for all tests was set at *p* < 0.05.

## Figures and Tables

**Figure 1 ijms-26-00775-f001:**
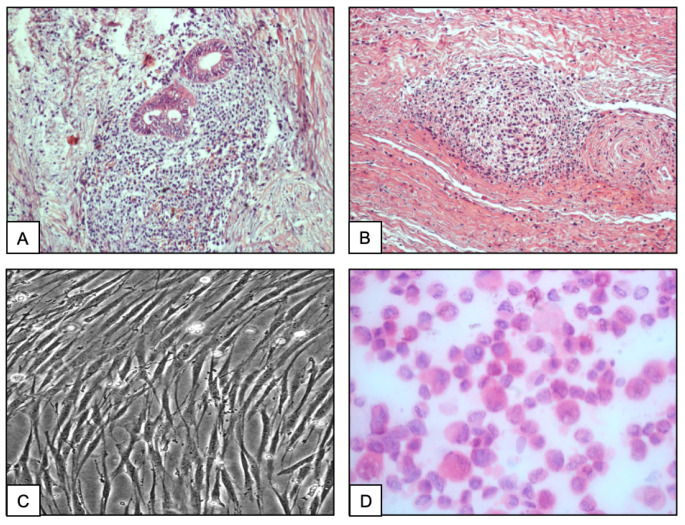
Histopathological aspect of an area of the AWE showing endometriotic glands surrounded by an abundant stroma composed of heterogeneous mesenchymal cells infiltrating the muscular tissue ((**A**) magnification ×20). Aspect of an isolated island composed exclusively of stromal cells implanted in the abdominal muscle ((**B**) magnification ×20). Phase contrast microscopy of adherent stromal cells grown in culture chambers (Labteks) ((**C**) magnification ×63). Stromal cell suspensions prepared as cytospins on slides. Note the change from polygonal cells to round morphology (H&E, magnification ×40) (**D**).

**Figure 2 ijms-26-00775-f002:**
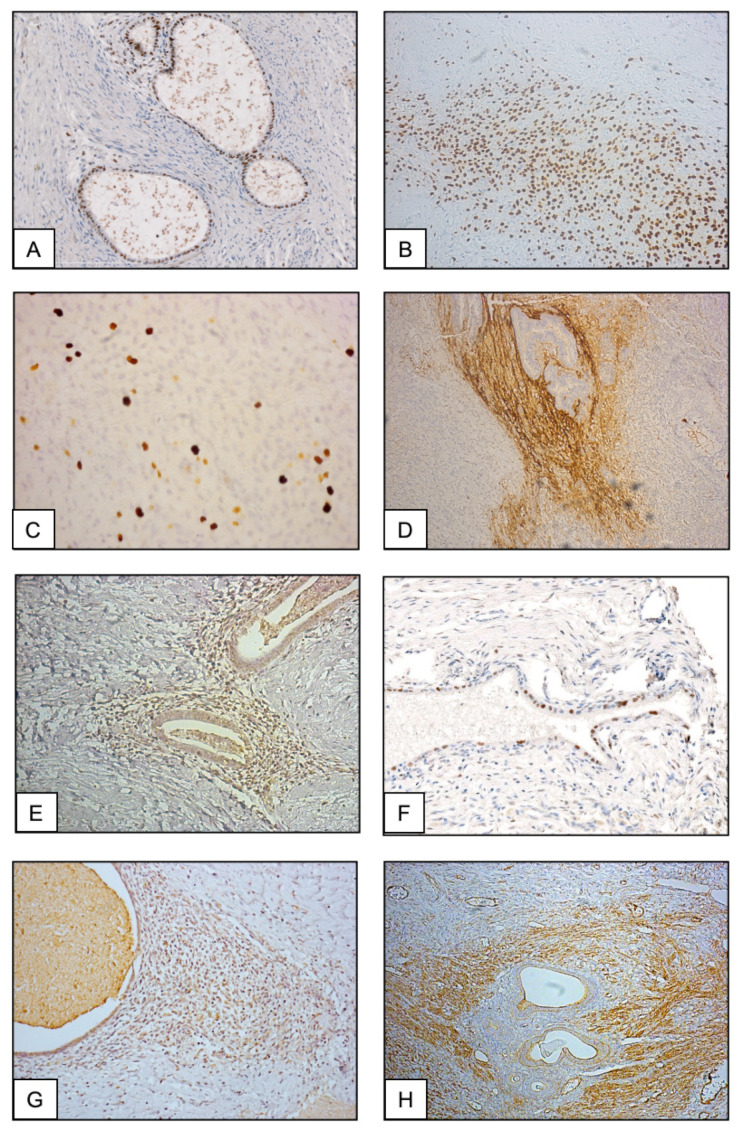
Representative immunostaining aspect of serial sections from the abdominal wall endometrioma with anti-Estrogen receptor ((**A**) magnification ×20), anti-Progesterone receptor PR ((**B**) magnification ×20), anti-Ki-67 ((**C**) magnification ×40), anti-c-MYC ((**D**) magnification ×20), anti-p16(INK4a) ((**E**) magnification ×20), anti-p53 ((**F**) magnification ×63), anti-CD10 ((**G**) magnification ×20) and anti-CD73 antibodies ((**H**) magnification ×20).

**Figure 3 ijms-26-00775-f003:**
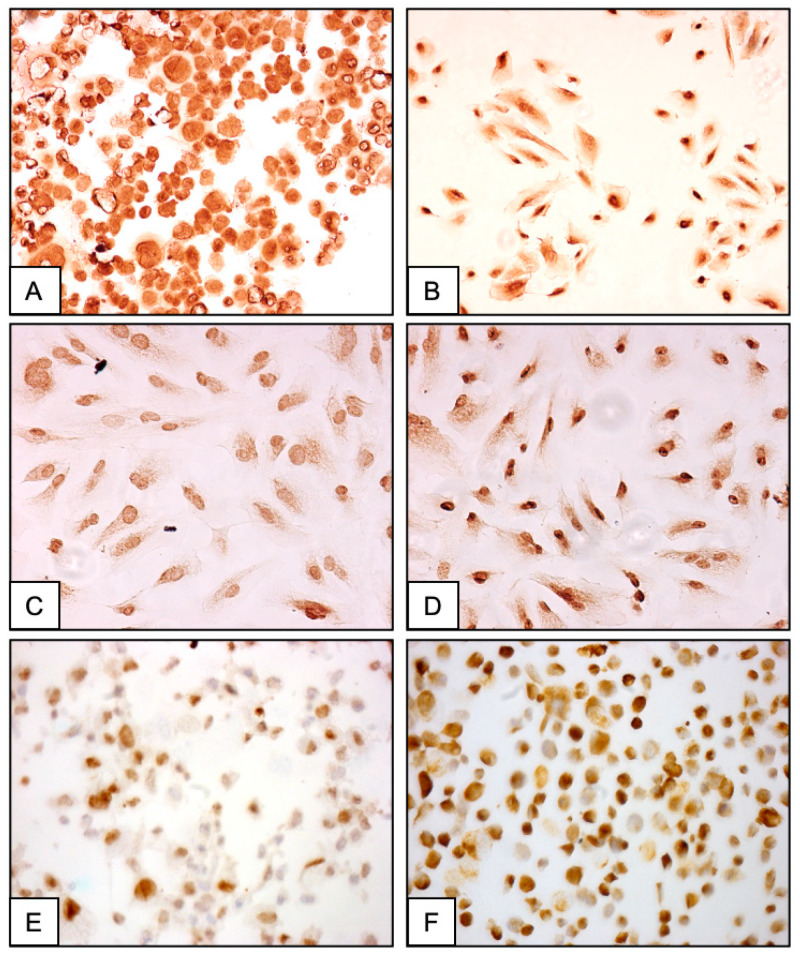
Representative immunostaining of the AWE-derived cultured stromal cells (magnification ×100). Prepared as cytospins and immunostained with anti-vimentin antibody (**A**). Adherent stromal cells immunolabeled with anti-c-MYC (**B**); anti-p16(INK4a) (**C**); anti-WT1 (**D**); anti-p53 (**E**) and anti-Rb (**F**) antibodies.

**Figure 4 ijms-26-00775-f004:**
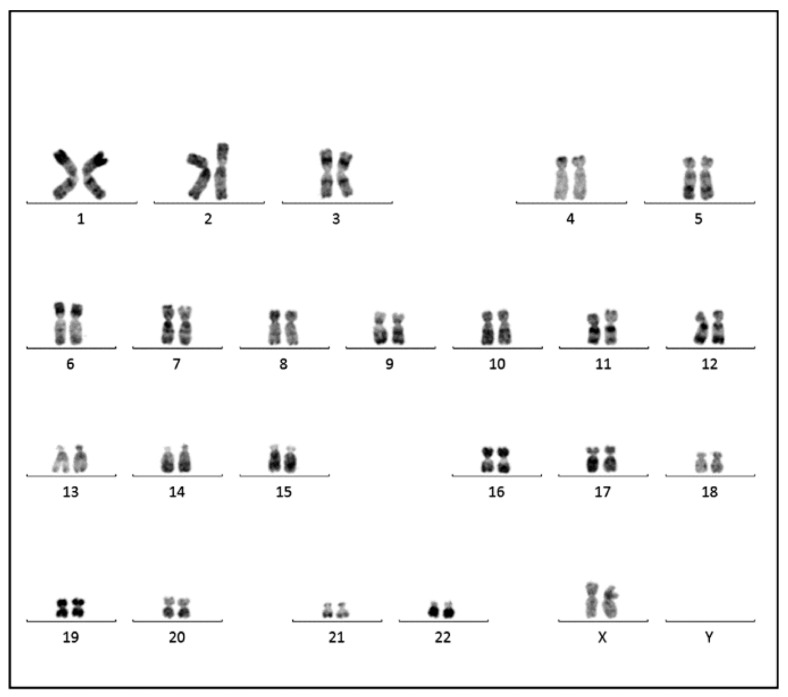
Conventional cytogenetic analysis of the cultured cells (metaphase spread, RHG-banding) from the fourth in vitro passage showing normal female karyotype (46XX). Presence of the normal chromosome 8q24 structure, the site of location of the c-MYC gene.

**Figure 5 ijms-26-00775-f005:**
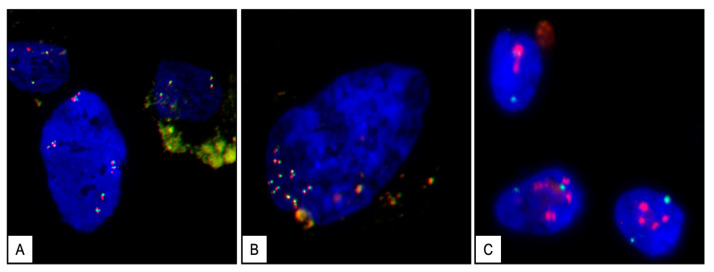
Fluorescence in situ hybridization (FISH) with MYC/CEN8 probes (DAPI counterstaining) showing different c-MYC-red/CEN8-green signal patterns observed in interphase nuclei of both tissue and cultured stromal cells. Aspect of an endometrioma nucleus showing several clusters of C-MYC/CEN8 signals interpreted as low level of amplification ((**A**) magnification ×63). Representative interphase nucleus of the endometrioma, showing more than 6 c-MYC/CEN8 signals ((**B**) magnification ×100). FISH assay applied to cultured stromal cells showing a high level of amplification of c-MYC gene distributed as large dispersed clusters (red) and two centromere CEN8 (green) signals ((**C**) magnification ×63).

**Table 1 ijms-26-00775-t001:** Immunohisto/cytochemical analysis of the abdominal wall endometrioma vs. cultured cells.

Antibody Used	Specificity	Source	Abdominal Wall Endometriosis	Mid Term Cell Culture
Cytokeratin (cl. AE1/AE3)	Epithelial cells	Dako-Agilent	Weak/moderate Gl.	2–3%
Vimentin (cl. V9)	Mesenchymal cells	Dako-Agilent	Negatif Gl., Moderate St.	94%
Ki67 (cl. Mib-1)	Proliferating cells	Dako-Agilent	Weak (5% of St.), Gl. = 0	15%
ER (cl. EP1)	Estrogen receptor	Dako-Agilent	Moderate/Strong Gl., St. = 0	45%
PR (cl. PgR 1294)	Progesterone receptor	Dako-Agilent	Strong St.	65%
CD9 (cl. 4H7B9)	Tetraspanin Cell adhesion molecule	ThermoFisher	ND	55%
CD10 (cl. DAK-CD10)	Common acute leukemia antigen	Dako-Agilent	Strong St.	85%
CD13 (cl. 3D8)	Aminopeptidase	Santa Cruz	ND	80%
CD73 (cl. EPR6114)	Mesenchymal stromal/stem cells *	Abcam	Weak Gl., St.	75%
CD90 (cl. 7E1B11)	Mesenchymal stromal/stem cells *	Abcam	Weak Gl., St.	68%
CD105 (cl. EPR10145-12)	Mesenchymal stromal/stem cells *	Abcam	Weak Gl., St.	61%
HLA-DR (cl. L243)	Class II MHC	Dako-Agilent	Weak Gl., St.	0
c-Myc (cl. C-33)	Proto oncogene	Santa cruz	Strong St., Weak Gl.	60%
P16 (INK4a) (cl. 1E12E10)	Cyclin-dependent kinase inhibitor	ThermoFisher	Strong St.	40%
P53 ^Wt-Mt^ (cl. DO-7)	P53 tumor suppressor	Dako-Agilent	Weak Gl., St.	32%
Rb (cl. 1F8)	Retinoblastoma protein	ThermoFisher	Weak Gl., St.	35%
WT1 (cl. 6F-H2)	Wilms tumor protein I	Dako-Agilent	Moderate Gl., St.	45–50%
CDL1 (cl. h-CALD)	Caldesmon	Santa Cruz	Moderate Gl., St.	65%

* Compared to Dominici et al. (antibodies) [18]. Gl.: Glands; St: (stroma); ND: Not Done; cl.: Clone.

## Data Availability

Data is contained within the article.

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
