# Peer review of "Idiopathic Abdominal Wall Endometrioma: Case Report with Investigation of the Pathological, Molecular Cytogenetic and Cell Growth Features In Vitro"

_ijms, 2025, doi:10.3390/ijms26020775_

Round 1

Reviewer 1 Report

Comments and Suggestions for Authors

Dear authors, I appreciate the effort and dedication that you have put into the manuscript and would like to offer my constructive feedback to help strengthen the quality of the work.

- since this is a case report, i think care guidelines should be included and it should also be stated (in the manuscript) that you used them

- line 186: form (FROM?) the disease arising. Please check other text for potential errors.

- is it possible to add some visuals regarding expression of oncoproteins etc.?

- what would be the limitations regarding your case and research? and what would be interesting to explore in the future regarding this topic? please explain and add

Author Response

Comments 1: since this is a case report, i think care guidelines should be included and it should also be stated (in the manuscript) that you used them.

Response 1: We agree with the reviewer, thus information has been added in the material and method section (4.1 subheading) as follows: The authors followed the care protocol guidelines approved by the ethics committee in our Institution and further obtained an informed consent for the patient privacy protection.

Comments 2: line 186: form (FROM?) the disease arising. Please check other text for potential errors.

Response 2: The sentence on lines 184 -187 was corrected as required, now reading; « So far, no specific histopathological, cytological or molecular features are recognized allowing to differentiate the primary abdominal wall endometriosis from  the disease arising in the abdominal surgical scars, generally described as cutaneous surgical endometriosis (2,9) ».

Comments 3: is it possible to add some visuals regarding expression of oncoproteins etc.?

Response 3: The presented data describing the expression of selected oncoproteins including c-myc, p53, Rb, p16, and WT1 are summarized in Table 1. If we follow the reviewer’s suggestion, creation of a new visuals would considerably increase the wording and space of the article, which is limited for a case report, (according to the IJMS instructions for authors). Thus, we are constrained to avoid creation of a new illustration figure.

Comments 4: what would be the limitations regarding your case and research? and what would be interesting to explore in the future regarding this topic? please explain and add

Response 4: The limitation of our study is mainly the examination of only of a single AWE tissue sample. This does not allow to propose comprehensive phenotypic and genotypic features of AWE that would delineate the specific form. Moreover, because of the rarity of AWE, collaborative multicentric research groups are desirable to  define more precisely the pathogenesis and predict the oncogenetic potential of these lesions.

Reviewer 2 Report

Comments and Suggestions for Authors

The manuscript by Gogusev and co-authors desribes pathological features of a case of the abdominal wall endometrioma. The study is novel and original, it addresses an understudied subset of endometriosis in which necrotizing endometrial tissue implants exclusively to the abdominal wall. The manuscript is well-written and well-illustrated. To improve readability of the manuscript, several critique points need to be addressed.

1. Table 1. Indicate the the catalog numbers of all antibodies used for the IHC.  In addtion, indicate what form of p53 (wild type or mutant) the corresponding antibody is designed to detect. 

2. It needs to be noted that WT1 is not a specific marker of endometriosis.

3. The authors should comment whether there is an established link between AWE and endometrioid subtype of epithelial ovarian carcinoma. What is known regarding the correlation between the implantation sites and endometrial ovarian carcinoma?

4. Please comment on an apparent discrepancy of low Ki67 and a concomitant high c-myc expression.

5. Please comment on a relatively low level of ER expression (Table 1). 

6. Sentences a) lines 91-92 and 2) lines 214-217 require corrections. 

Author Response

Comments 1: Table 1. Indicate the catalog numbers of all antibodies used for the IHC.  In addition, indicate what form of p53 (wild type or mutant) the corresponding antibody is designed to detect. 

Response 1: As asked by the reviewer, the catalog numbers of all antibodies and/or clones are now presented in Table 1. We confirm  in the table that  the p53 antibody used, recognise both the wild-type and the mutant form of the molecule. Moreover, we have included in the table the name of the clone for each antibody used.

Comments 2: It needs to be noted that WT1 is not a specific marker of endometriosis.

Response 2: We agree with the reviewer that WT-1 is a tumour suppressor protein (tissue specific developmental regulator) and is also expressed in various malignancies including ovarian epithelial and stromal tumours. However, its expression is rarely investigated in endometriosis. We show here its expression in the studied case of AWE, although the WT-1 is not a specific marker of endometriosis.   

Comments 3: The authors should comment whether there is an established link between AWE and endometrioid subtype of epithelial ovarian carcinoma. What is known regarding the correlation between the implantation sites and endometrial ovarian carcinoma?

Response 3: We added in the text the following sentence “In the same context, several studies have reported epidemiologic and histopathological association between endometriosis and ovarian neoplastic processes.Yet the molecular and cellular mechanisms remain not well understood. In this regard, the  genetic link has been suggested through mutations in PIK3CA, KRAS, ARIDIA1A and other genes (Bulun, SE.; Wan, Y.; Matei, D. Epithelial Mutations in Endometriosis: Link to Ovarian Cancer. Endocrinology, 2029, 160, 626-638. DOI:10.1210/en.2018-00794).

Little is known concerning the correlation between the implantation sites and endometrial related carcinoma. Future studies are needed to establish or not, the link between these pathologies.

Comments 4: Please comment on an apparent discrepancy of low Ki67 and a concomitant high c-myc expression.

Response 4: We repeatedly examined our results concerning the expression of Ki-67 values on tissue and the cell culture experiments. An unintentional error was done typewriting the percentage values on the Table 1. So now, a corrected mean value of 15% for Ki-67 is reported in the Table 1. The discrepancy in the expression of Ki-67 and c-myc  proteins, observed in the tissue and in cell culture might be explained by the c-myc gene rearrangement (amplification) observed by the FISH experiments.

Comments 5: Please comment on a relatively low level of ER expression (Table 1). 

Response 5: The expession of ER was evaluated by the Image J, and QUPath softwares of randomly (blindly) chosen areas from the immunostained slides evaluating the intensity and the persentage of the staining. A strong ER expression was observed only in the endometriotic glands lining cells, whereas a lower percentage of stromal cells were  weakly labeled.Since the ER positive glandular component was heterogeneously scattered through the lesion, the mean value of staining for both cell types was evaluated.  A value of approximately 45% of ER positivity was obtained. compared to the stromal component. Threfore, we couldn’t compare ER  expression in tissue and the expression in the cell culture; the glands/stroma architecture is heterogeneous.

Comments 6: Sentences a) lines 91-92 and 2) lines 214-217 require corrections. 

Response 6: We corrected  the sentence on lines 91-92;  “After several passages the cells and started a continuous growth as an elongated large mesenchymal-like cells with a doubling time between 5 and 6 days.” has been replaced by “After several passages the cells started a continuous growth as an elongated large mesenchymal-like cells with a doubling time between 5 and 6 days.”

The sentence on lines 214-217, « Similarly, two different malignancies arising in cutaneous umbilical endometriomas. histologically described as clear cell and serous adenocarcinomas were linked to alter protooncogenes gene structures and aberrant mesenchymal stromal cell proteins expression [39–41]. » has been replaced by« Similarly, two different malignancies arising in cutaneous umbilical endometriomas. histologically described as clear cell and serous adenocarcinomas were linked to altered protooncogenes structures and aberrant mesenchymal stromal cell proteins expression [39–41]. »